# Effects of Alterations in Acid–Base Effects on Insulin Signaling

**DOI:** 10.3390/ijms25052739

**Published:** 2024-02-27

**Authors:** Lynda A. Frassetto, Umesh Masharani

**Affiliations:** 1Department of Medicine, Division of Nephrology, University of California San Francisco, San Francisco, CA 94143, USA; 2Department of Medicine, Division of Endocrinology, University of California San Francisco, San Francisco, CA 94143, USA

**Keywords:** acid stress, hormone, growth, lipolysis, proteolysis, renal function

## Abstract

Insulin tightly regulates glucose levels within a narrow range through its action on muscle, adipose tissue and the liver. The activation of insulin receptors activates multiple intracellular pathways with different functions. Another tightly regulated complex system in the body is acid–base balance. Metabolic acidosis, defined as a blood pH < 7.35 and serum bicarbonate < 22 mmol/L, has clear pathophysiologic consequences including an effect on insulin action. With the ongoing intake of typical acid-producing Western diets and the age-related decline in renal function, there is an increase in acid levels within the range considered to be normal. This modest increase in acidosis is referred to as “acid stress” and it may have some pathophysiological consequences. In this article, we discuss the effects of acid stress on insulin actions in different tissues.

## 1. Introduction

### Regulation of Acid–Base Balance

One of the most tightly controlled systems in the body is the amount of free hydrogen ions [H+] available, which is given by the equation
[H^+^] = PCO_2_/[HCO_3_^−^].(1)

In order to maintain the concentration of free hydrogen ions within the normal human range of 35–45 nmol/L, the lungs and the kidneys work together to excrete carbonic acid as carbon dioxide (CO_2_) and non-carbonic acids as ammonium, sulfate, phosphate, chloride and organic acids [1]. In addition, there are multiple systems that both buffer free hydrogen ions, for example, by exchanging them for potassium in red blood cells, and neutralize them, for example, via the dissolution of the bone matrix to release basic carbonate salts (bicarbonate, HCO_3_^−^) [1,2].

The inability to maintain systemic acid levels within the normal range can occur with excessive dietary acid intake or increased endogenous acid production, inadequate acid excretion, inadequate acid neutralization or intracellular transport [2]. Because the lungs have a very high capacity to excrete carbon dioxide, chronic respiratory acidosis only occurs in subjects with advanced damage to their alveoli, and will not be discussed here. Instead, we will discuss metabolic acidosis (defined as a blood pH < 7.35 and serum bicarbonate < 22) in subjects with advanced renal disease, and/or “acid stress”, if the free hydrogen ion levels are chronically on the high side of normal (e.g., 41–45, corresponding to a blood pH of 7.39–7.35, respectively) [3].

Acute acid–base imbalances can be caused by both electrolyte disorders, such as the effects of hypokalemia on the distal renal tubule collecting duct, which increases urinary hydrogen ion excretion [4], and acute illnesses such as sepsis-induced hypotension and inflammation [5].

Long-term or chronic acid stress and metabolic acidosis can lead to pathophysiologic consequences, many of which are covered in this Special Issue. In this chapter, we will discuss insulin signaling in various tissues, and the effects of acid stress and bicarbonate treatments on these body systems.

## 2. Chronic Acid Stress and Metabolic Acidosis Develop with Increasing Age and Renal Functional Decline

As we age, our renal function, on average, declines about 1 mL/min/year so that, by 70 years old, approximately half of the renal function one has at birth is gone. This is in part due to glomerular sclerosis and subsequent renal tubular dropout [6]. For acid–base balance, both the glomeruli and the renal tubules are important to filter phosphates, sulfates and chlorides; to secrete organic acids; to excrete hydrogen ions, both as ammonium (NH4) and directly via transporters in the distal tubules; and to reabsorb bicarbonate [1].

The loss of nephron function with aging has been associated with subtly lowering blood pH levels and decreasing serum bicarbonate levels; i.e., increasing acid stress. In an analysis of 23 articles comprising 971 individuals, Frassetto and Sebastian [7] demonstrated that, from 20 to 80 years of age, participants’ blood pH decreased from 7.41 to 7.38 (*p* < 0.001) and their serum bicarbonate decreased from 26.2 mmol/L to 22 mmol/L (*p* < 0.001). This was associated with an expected compensatory decrease in PCO2 (*p* < 0.005). These authors further demonstrated, in 64 individuals aged 17–74 years old that were admitted to the UCSF General Clinical Research Center for metabolic balance studies, that the blood’s hydrogen ion content increases with increasing PCO_2_, with renal net acid excretion [NAE, defined as the sum of the urine ammonium (NH_4_) and titratable acid minus the urine bicarbonate concentration] (*p* < 0.001), with age (*p* < 0.002), and with declining renal function (glomerular filtration rate, GFR; *p* < 0.001). Participants’ bicarbonate levels decreased with increasing age (*p* < 0.001) and NAE (*p* < 0.001) and decreasing GFR (*p* < 0.001) [8].

The effects of metabolic acidosis in subjects with advanced renal disease (generally when their GFR falls below 30 mL/min) have been described in some detail. These include bone breakdown [9], muscle proteolysis [10] and a more rapid loss of renal function [11]. This is in part due to metabolic acidosis’ direct physico-chemical dissolution of bone and indirect effects on bone via the activation of osteoclasts; via paracrine effects, as in upregulation of renin-angiotensin and endothelin levels, both of which affect renal function; and via its effects on cell membrane transporter systems, such as insulin and insulin-like growth factor 1 (IGF1) in muscle. Insulin signaling will be discussed in more detail in the next section.

## 3. Insulin Receptors and Insulin Signaling: Its Very Broad Effects in Tissues Besides Glucose Uptake

Insulin was first isolated from pancreatic extracts more than 100 years ago, and since then a great deal of research has been carried out on insulin actions [12]. Insulin acts by binding to its membrane receptor. The receptor consists of two extracellular alpha subunits and two transmembrane beta subunits. The beta subunit is a tyrosine kinase. The binding of insulin to the extracellular alpha subunit induces conformational changes in the beta subunit and activates its kinase function. The activated kinase then acts on intracellular signaling proteins (Figure 1). The phosphatidylinositol 3 kinase (PI3K) signaling cascade is responsible for the metabolic responses regulated by insulin. The RAS-MAP kinase signaling cascade activates genes for cell growth, division and differentiation [13,14].

Insulin’s principal action is to regulate and maintain glucose levels within a narrow range through its effect on muscle, adipose tissue and the liver [14]. Insulin is released from beta cells in response to an increase in blood glucose levels. Its actions include the suppression of glycogenolysis, gluconeogenesis, ketogenesis, lipolysis and proteolysis ([15], Figure 2). The consequence of these actions is to promote glucose utilization. The tissue-specific inactivation of insulin receptors in experimental systems provides important insights into the action of insulin on intermediate metabolism [16,17,18,19].

### 3.1. Liver

In the liver, insulin action promotes glycogen synthesis, glycolysis and lipogenesis. The insulin activation of phosphoinositide 3-kinase (PI3K)/Akt is critical for many of these actions.

Akt increases the expression of sterol regulatory element binding protein 1c (SREBP1c), the master transcriptional regulator of lipogenic enzymes. The expression of SREBP1c and lipogenic enzymes promotes fatty acid production as well as triglyceride esterification and secretion [20].

Liver glycogen stores are essential for maintaining glucose homeostasis during fasting. Although insulin does not regulate the uptake rate of hepatic glucose (unlike in muscle tissue), insulin is required for glycogen synthesis. Multiple insulin-regulated enzymes participate in glycogen synthesis, including in activating the gene glucokinase, a key regulator of glycogenic synthetic flux [14].

Fatty acids are broken down into acetyl-CoA in the mitochondria. The increased availability of acetyl-CoA is important in the production of ketones; high ketone levels inhibit the glycolytic pathway, suppressing glucose utilization [16]. Insulin suppresses ketogenesis by limiting the entrance of fatty acids into mitochondria for the generation of acetyl-CoA.

Insulin inhibits hepatic gluconeogenesis through both direct and indirect effects. Its indirect effects include the inhibition of lipolysis and decreased protein catabolism in muscle. Insulin action via Akt and the transcription factor forkhead box protein O1 (FOXO1) inactivates glycogen synthase kinase 3β (GSK3) and reduces the expression of gluconeogenesis genes including pyruvate carboxylase, glucose-6-phosphatase and phosphoenolpyruvate carboxykinase [15].

A liver-specific insulin receptor knockout mouse (LIRKO) had elevated insulin levels, impaired glucose intolerance and an inability to suppress its hepatic glucose production [21]. The mouse also has reduced serum triglycerides, increased fatty acid oxidation and an inability to induce de novo lipogenesis in response to a meal. The disruption of FOXO1 in the LIRKO mouse normalizes its hepatic glucose production, indicating that insulin can effectively regulate hepatic glucose production via indirect pathways [21].

### 3.2. Adipose Tissue

Insulin’s principal action on the adipocyte is to limit lipolysis by inhibiting hormone-sensitive lipase. Insulin increases the glucose uptake in adipocytes but this drives a relatively small fraction of de novo lipogenesis, with the majority of the process accounted for by the esterification of preformed fatty acids. The carbohydrate response element binding protein (ChREBP) plays a major role in adipocyte lipogenesis [22].

Insulin lowers cyclic AMP (cAMP) levels, which in turn reduce the activity of cAMP-dependent protein kinase (PKA) and hormone-sensitive lipase. Perilipin is a substrate of PKA and regulates the access of cytosolic lipases to lipid droplets [23]. Insulin is an important phosphoprotein which coats lipid droplets. Perilipin activation inhibits the first step of the hydrolysis of triglycerides to diacylglycerols. Perilipin and PKA therefore both target hormone-sensitive lipase of the surface of lipid droplets. In this way, insulin, due to its effects on PKA, periplipin and hormone-sensitive lipase, suppresses lipolysis [23].

Bluther et al. studied the role of insulin signaling in a fat-cell-specific insulin receptor knockout mouse (FIRKO) model. With aging, FIRKO mice had a lower fat mass and lipo-dystrophy compared to controls (*p* < 0.05), but normal fasting glucose and normal glucose tolerance tests, which were maintained even with aging [18]. This model shows the im-portance of insulin signaling in triglyceride storage.

### 3.3. Muscle

In muscle tissue, insulin action stimulates glucose uptake and glycogen synthesis. Insulin increases glucose transport into the myocytes by recruiting glucose transporter type 4 (GLUT4) transporters to the plasma membrane. This process is thought to be under PI3K-dependent control. The increased substrate availability within the myocytes drives glucose oxidation and upregulates enzymes such as phosphorylated glycogen synthetase to help promote glycogen storage [14].

Insulin also inhibits muscle protein breakdown, reducing the availability of gluconeogenic precursors [24]. A muscle-specific insulin receptor knockout mouse (MIRKO) had, as might be expected, a marked decrease in the ability of its insulin to stimulate glucose transport into its muscles. This, however, did not result in elevated systemic insulin levels or elevated fasting glucose levels. Its glucose tolerance was also normal. There was a modest increase in its fat mass and it had elevated free fatty acid levels and serum triglycerides. Its muscle mass was normal and there was no impact on its spontaneous activity or ability to run on a treadmill compared to control littermates.

### 3.4. Central Nervous System

Insulin receptors are widely expressed in the central nervous system in neurons and glial cells. Insulin enters the central nervous system principally via receptor-mediated transcytosis. Insulin action in the brain may be important in the regulation of feeding behavior [25]. An intracerebroventricular infusion of insulin decreased the food intake in baboons [26]. A central nervous system-specific insulin receptor knockout mouse (NIRKO) displayed diet-sensitive obesity [17].

### 3.5. Growth

Epidemiological studies have linked hyperinsulinemia to an increased risk of breast, endometrial, ovarian and prostate cancer [27]. The mechanisms of this link are not well defined, and several mechanisms have been proposed [28,29]. First, insulin action on the insulin receptor activates the RAS-MAP kinase signaling cascade, promoting growth and proliferation (Figure 1). Second, insulin can promote an increased production of IGF1 by the liver. Third, the insulin receptor (IR) and the type 1 insulin-like growth factor receptor (IGF-1R) are closely related receptors. In cells that express both receptors, IR and IGF-1R alpha beta monomers can form functional hybrid IR/IGF1R receptors. Insulin, IGF-I and IGF-II are all capable of binding the various hybrid receptors [28]. Targeting these IR/IGF1R receptors has been a focus of recent cancer treatments [29].

## 4. Evidence of Acid Stress and Metabolic Acidosis’ Effects on Insulin Action

In this section, we will discuss the evidence of the effects of acid stress on insulin signaling and insulin action. In vitro studies of the effects of pH on insulin binding in cultured human leukocytes demonstrate a U-shaped curve, with decreased binding/dissociation rates at both low (pH 6.7) and high (pH 9.0) pH levels, with increased activity in the normal human pH range (7.35–7.45) [30].

### 4.1. Kidney

Insulin resistance occurs in patients with moderate to severe uremia. Compared to controls with normal renal function, non-diabetic patients on dialysis, while acidotic, were twice as insulin-resistant. Two weeks of treatment with sufficient sodium bicarbonate to neutralize their metabolic acidosis resulted in a significant improvement in their insulin sensitivity, but it did not normalize their pH [31].

DeFronzo studied glucose metabolism by inducing metabolic acidosis in healthy, normal-weight volunteers. The ingestion of ammonium chloride for three days lowered their blood pH from 7.41 ± 0.01 to 7.37 ± 0.02, while their serum bicarbonate decreased from 23 ± 1 to 19 ± 1 mmol/liter (*p* < 0.001). The results of a hyperinsulinemic euglycemic clamp demonstrated that their glucose uptake decreased from 7.63 ± 0.63 to 6.42 ± 0.49 mg/kg body wt per min (*p* < 0.005) and that their M/I ratio (a measure of the quantity of glucose metabolized per unit of plasma insulin concentration) decreased from 6.61 ± 0.45 to 5.49 ± 0.35 (*p* = 0.01), indicating a decline in insulin sensitivity of about 15% [32]. Their hepatic sensitivity to insulin action, however, was not altered.

The observation that acidosis increases insulin resistance may explain why a high dietary acid load increases the risk of type 2 diabetes [33]. In a study of 66,485 women, the quartile with the highest dietary acid load was associated with a significant risk of type 2 diabetes. The association was particularly notable in women with a BMI < 25 kg/m^2^ (hazard ratio 1.96, 95% CI 1.43, 2.69) [33].

### 4.2. Muscle

The mechanism by which declines in pH affect glucose disposal remains uncertain. In studies of the effect of pH on exercise, normal volunteers underwent progressive exercise tests on a cycle ergometer following acute bicarbonate-induced alkalosis or ammonium chloride-induced acidosis. Exercise endurance at 95% maximum oxygen uptake (VO_2_ max) was the longest with alkalosis and the shortest with acidosis. Individuals with acidosis appear to have impaired muscle glycolysis and reduced lactate efflux from their muscle [34,35]. Measurements of glycolytic intermediates in muscle biopsies performed during exercise found that acidosis was associated with decreases in the concentrations of glycolytic intermediates from fructose-1,6-bisphosphate to phosphoenol pyruvate. This patten of muscle glycolytic intermediates’ concentrations suggests that, in acidosis, phospofructokinase (PFK) activity is inhibited. Phosphofructokinase catalyzes the phosphorylation of fructose-6-phosphate to fructose-1,6-bisphosphate and is a key regulatory enzyme in the sequential conversion of glucose to lactate. PFK is also very sensitive to pH changes. Decreases in blood pH (and serum bicarbonate concentration) may prevent the binding of ATP to PFK, inhibiting enzyme action, and, thus, may play a key role in the pH control of glycolysis [36].

### 4.3. Adipose

Severe acidosis could also directly affect insulin action at the adipocyte receptor. Whittaker et al., in a rat adipose model, studied the change in insulin binding in rats with severe ammonium chloride-induced acidemia (blood pH 6.72 ± 0.04 vs. normal controls). The acidotic rats had higher blood glucose levels as well as increased blood insulin levels, suggesting that the acidosis reduced insulin binding. The authors suggested that this could result from a reduced insulin receptor concentration [37], while Waelbroecke demonstrated that insulin binding decreases at lower pHs [30].

### 4.4. Liver

In vitro experiments in isolated perfused rat livers have shown that hepatic gluconeogenesis is markedly inhibited (by almost 10-fold) when the pH of the perfusate is less than 7.1 [38]. There was, however, no alteration in gluconeogenesis for pH changes between 7.1 and 7.6. During hydrochloric acid-induced metabolic acidosis in anesthetized dogs, hepatic glucose production decreased by approximately 30% (*p* < 0.005), which the authors suggested might be due to a decrease in glycogenolysis [39].

### 4.5. Central Nervous System

Insulin is the key factor to maintaining low levels of acid-sensing ion channels (ASICs) on the plasma membrane. ASICs are voltage-independent sodium channels activated by extracellular protons. In cerebral ischemia-induced low blood flow states, the combination of acidosis and decreases in serum insulin levels results in increased ASIC activity, and subsequent greater tissue infarction [40]. Whether or not a reversal of the acidosis results in a decreased infarct size is under investigation (UCSF B. Lindquist, personal communication).

## 5. Evidence That Bicarbonate Treatment Improves Insulin Signaling

If acid stress and metabolic acidosis are factors that influence insulin signaling and downstream metabolic effects, then, potentially, an alkali treatment, neutralizing the effects of acidosis, would alter or help normalize the effects of that acidosis. The effects of alkali therapy on slowing the decline in renal function in chronic kidney disease (CKD) have been recently discussed [41,42].

Bellasi et al. [43], in 145 subjects with well-controlled diabetes [glycosylated hemoglobin (HgA1c) average 6.7] and CKD 3b-4 (GFR 20–45 mL/min), demonstrated that normalizing serum bicarbonate levels by giving subjects an average of 0.7 ± 0.2 mmol/kg of bicarbonate daily resulted in improved glucose control by (1) lowering their insulin levels (13.4 ± 5.2 vs. 19.9 ± 6.3 for treated vs. control subjects, respectively; *p* < 0.001), (2) decreasing their HOMA-IR (Homeostatic Model Assessment for Insulin Resistance (5.9 [5.0–7.0] vs. 6.3 [5.3–8.2]; *p* = 0.01) and (3) decreasing the need for oral antidiabetic drugs. The largest HOMA-IR reduction was noted for serum bicarbonate levels between 24 and 28 mmol/L.

Reaich et al. [44] studied the effects of acidosis on insulin-mediated protein degradation. Eight non-diabetic men, aged 20–70 years old with CKD 5 (GFR < 15 mL/min, not yet on dialysis), with an average blood pH of 7.29 ± 0.01 and serum bicarbonate 17 ± 1 mmoL/L, were studied before and after 4 weeks of treatment with oral bicarbonate, 0.85 ± 0.06 g/kg/day. Protein degradation was determined by the kinetic changes in L [1-^13^C] leucine infusion during a hyperinsulinemic euglycemic clamp. After the bicarbonate treatment, their mean blood pH rose to 7.36 ± 0.01 (*p* < 0.001) and their serum bicarbonate rose to 20 ± 1 mmol/L (*p* < 0.001). Insulin sensitivity, measured as the glucose infusion rate/measured insulin level, increased from 6.4 ± 1.2 to 7.3 ± 1.1 (*p* < 0.05). Protein degradation decreased during the clamp both before and after bicarbonate treatment.

An alternative to bicarbonate supplementation is eating a diet high in fruits and vegetables, which contain alkali precursors metabolizable to bicarbonate [45]. Eating a diet high in fruits and vegetables that avoids processed foods can also help improve glucose control and insulin levels. In the WHEL multicenter trial, more than 3000 women previously treated for breast cancer in the United States were randomized to a healthy diet or a high-carotene-containing fruit and vegetables diet. Those with the lowest intake of fruits and vegetables (which contain organic anions metabolizable to bicarbonate, such as citrate and malate) had the highest diet acid load (DAL). In a multivariate adjusted model, compared to the women in the lowest quartile, those in the highest DAL quartile had the highest CRP (30–33%) and highest levels of HgA1c (6–9%) [46]. Similarly, in a study of 135 children and adolescents, those with the lowest DAL (as assessed by 3-day food records and dietary estimations of their DAL) had the lowest HOMA-IR [47].

Excessive bicarbonate treatments, especially when combined with renal insufficiency (because the kidneys cannot excrete enough bicarbonate), can lead to metabolic alkalosis and subsequent hypokalemia and urinary chloride loss [48]. As discussed earlier, insulin binding/dissociation rates are pH-dependent, and their rates increase as the pH rises [30]. A more in-depth review of bicarbonate effects is available in Kalani [49].

## 6. Conclusions

Insulin has a critical regulatory role in intermediate metabolism, regulating not only glucose levels but also proteolysis and lipolysis. Insulin is also important in cell growth and proliferation. Acidosis, due to its effects on insulin-regulated pathways, may increase proteolysis, lipolysis and inhibit cell growth and proliferation.

In as much as increased acid levels are an abnormal stress on the system, the neutralization of excess acid may be a key to reversing some of the ongoing pathophysiological effects induced by alterations in the regulation of insulin signaling.

## Figures and Tables

**Figure 1 ijms-25-02739-f001:**
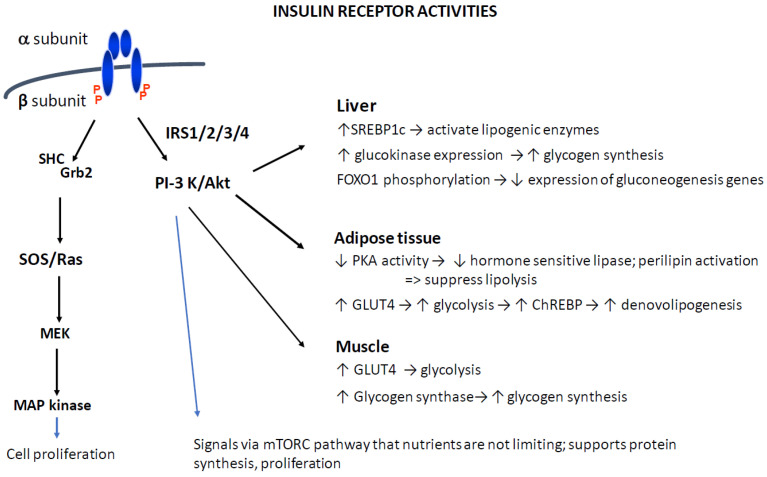
Intracellular insulin signaling pathways. Abbreviations: IRS—insulin receptor substrate; PI3K—phosphatidylinositol kinase; PDK3—phosphoinositide-dependent protein kinase 1; Akt—a seronine/threonine kinase, also known as protein kinases B; ChREBP—carbohydrate response element-binding protein; mTOR—mechanistic target of rapamycin; Grb2—growth factor receptor bound protein 2; MEK—mitogen activated protein kinase; MAP kinase—mitogen activated protein kinase.

**Figure 2 ijms-25-02739-f002:**
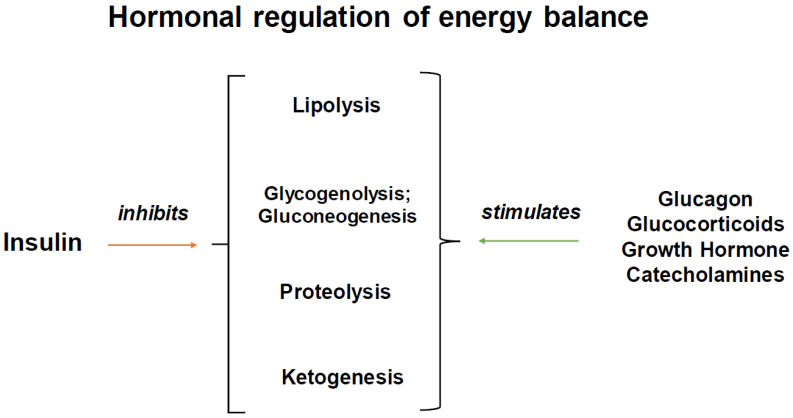
Effects of insulin and counterregulatory hormones on energy balance.

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
