# Peer review of "Effects of Alterations in Acid–Base Effects on Insulin Signaling"

_ijms, 2024, doi:10.3390/ijms25052739_

Round 1

Reviewer 1 Report

Comments and Suggestions for Authors

This succinct review addresses the effect of dysregulation of the acid-base balance such as metabolic acidosis or so-called “acid stress” on insulin signaling and action. The topic is interesting and the review is well-written, but in my view is somewhat superficial (only 38 references) and could be improved.

Specific comments

Introduction and section 2: multiple statements in lines 19-49 completely lack references.

Figures 1 and 2 are oversimplistic and somewhat incorrect. The authors can surely find more complete and accurate figures describing insulin action and its physiological role in the literature e.g. refs. 6 and 7.

The statement on line 83 (and Figure 2) that insulin actions are principally inhibitory is blatantly incorrect. Insulin has a major role in stimulating glucose transport through membrane translocation of the glucose transporter GLUT4 in muscle and fat, in stimulating glycogen synthesis in the liver (mentioned in line 99 but not in Fig. 2), in stimulating lipogenesis in adipocytes, in stimulating amino acid uptake and protein synthesis, gene expression and DNA synthesis, cell growth and mitogenesis.

In fact, stimulation by insulin of glucose uptake in enzymatically isolated adipocytes (Rodbell M, J Biol Chem 239,375-380, 1964) using glucose-U-14C or 3-O-[methyl-3H]-glucose, and stimulation of lipogenesis in the same isolated adipocytes by measuring the incorporation of D-[3-3H]-glucose into lipids (Moody AJ et al, Horm Metab Res 6:12-16,1974) have been used for decades as a sensitive in vitro bioassay of insulin action. The authors describe the action of insulin in adipocytes as limited to inhibition of lipolysis.

The authors mention in line 188 that acidosis could also directly affect insulin action at the receptor level. They seem to be unaware of the extensive early documentation of the pH dependence of insulin-receptor binding affinity and kinetics with a bell-shaped curve that has been quantitatively analysed, showing among other things that acidification accelerates the dissociation of insulin from the receptor (as in the endosome after internalization). See for example Waelbroeck M, J Biol Chem 257:8289-8291,1982.

Line 149: refs. 17-19 are incorrect, they have nothing to do with acid stress.

Line 141: needs reference.

Author Response

We would like to thank both reviewers for their comments, which have contributed to improving our manuscript.

Reviewer #1:

This succinct review addresses the effect of dysregulation of the acid-base balance such as metabolic acidosis or so-called “acid stress” on insulin signaling and action. The topic is interesting and the review is well-written, but in my view is somewhat superficial (only 38 references) and could be improved.

Specific comments:

Introduction and section 2: multiple statements in lines 19-49 completely lack references.

We have added references to this section.

Figures 1 and 2 are oversimplistic and somewhat incorrect. The authors can surely find more complete and accurate figures describing insulin action and its physiological role in the literature e.g. refs. 6 and 7.

We have revised the figures to more completely demonstrate the regulatory role of insulin.  We do believe that complicated figures can be hard to interpret, and tried to therefore outline the important insulin actions.

The statement on line 83 (and Figure 2) that insulin actions are principally inhibitory is blatantly incorrect. Insulin has a major role in stimulating glucose transport through membrane translocation of the glucose transporter GLUT4 in muscle and fat, in stimulating glycogen synthesis in the liver (mentioned in line 99 but not in Fig. 2), in stimulating lipogenesis in adipocytes, in stimulating amino acid uptake and protein synthesis, gene expression and DNA synthesis, cell growth and mitogenesis.

We have revised the manuscript to better describe the regulatory role of insulin, i.e., both inhibitory and stimulatory effects.

In fact, stimulation by insulin of glucose uptake in enzymatically isolated adipocytes (Rodbell M, J Biol Chem 239,375-380, 1964) using glucose-U-14C or 3-O-[methyl-3H]-glucose, and stimulation of lipogenesis in the same isolated adipocytes by measuring the incorporation of D-[3-3H]-glucose into lipids (Moody AJ et al, Horm Metab Res 6:12-16,1974) have been used for decades as a sensitive in vitro bioassay of insulin action. The authors describe the action of insulin in adipocytes as limited to inhibition of lipolysis.

We thank the reviewer for these references.

The authors mention in line 188 that acidosis could also directly affect insulin action at the receptor level. They seem to be unaware of the extensive early documentation of the pH dependence of insulin-receptor binding affinity and kinetics with a bell-shaped curve that has been quantitatively analysed, showing among other things that acidification accelerates the dissociation of insulin from the receptor (as in the endosome after internalization). See for example Waelbroeck M, J Biol Chem 257:8289-8291,1982.

We thank the reviewer for this reference and recommendation.

Line 149: refs. 17-19 are incorrect, they have nothing to do with acid stress.

The reviewer is correct.  We have deleted these.

Line 141: needs reference.

We have added references.

Reviewer 2 Report

Comments and Suggestions for Authors

Lynda A. Frassetto and Umesh Masharani conducted a comprehensive review on the impact of acid stress on insulin action. It is widely acknowledged that the body's acid-base balance is integral to maintaining physiological homeostasis, and deviations from this equilibrium can significantly affect diverse cellular processes, including insulin signaling. However, a critical evaluation of this review reveals certain limitations, encompassing both the depth of molecular mechanisms and the clinical implications discussed.

The compilation of various reports in this review seems to present a somewhat superficial examination of the topic, lacking a thorough exploration of underlying molecular mechanisms. Additionally, the review overlooks critical aspects such as inflammation and electrolyte imbalances, which are pertinent factors influencing the intricate interplay between acid-base balance and insulin signaling.

Moreover, the review falls short in providing a comprehensive overview of the clinical implications associated with alterations in acid-base balance, leaving certain important dimensions unexplored. Notably, the authors missed addressing potential side effects related to bicarbonate therapy, a relevant aspect given its role in correcting acidosis. A more thorough consideration of these aspects would have strengthened the overall depth and applicability of the review, providing a more nuanced understanding of the intricate relationship between acid stress, insulin action, and therapeutic interventions such as bicarbonate therapy.

Comments on the Quality of English Language

Minor corrections are needed

Author Response

We would like to thank both reviewers for their comments, which have contributed to improving our manuscript.

Reviewer #2:

Lynda A. Frassetto and Umesh Masharani conducted a comprehensive review on the impact of acid stress on insulin action. It is widely acknowledged that the body's acid-base balance is integral to maintaining physiological homeostasis, and deviations from this equilibrium can significantly affect diverse cellular processes, including insulin signaling. However, a critical evaluation of this review reveals certain limitations, encompassing both the depth of molecular mechanisms and the clinical implications discussed.

The compilation of various reports in this review seems to present a somewhat superficial examination of the topic, lacking a thorough exploration of underlying molecular mechanisms.

We have added to our sections on molecular mechanisms.

Additionally, the review overlooks critical aspects such as inflammation and electrolyte imbalances, which are pertinent factors influencing the intricate interplay between acid-base balance and insulin signaling.

We have added more information about how inflammation and electrolyte imbalances can secondarily affect acid-base status.  However, we believe a detailed discussion of the mechanisms by which electrolyte imbalances or inflammation affect acid-base status is beyond the scope of this paper.

Moreover, the review falls short in providing a comprehensive overview of the clinical implications associated with alterations in acid-base balance, leaving certain important dimensions unexplored. Notably, the authors missed addressing potential side effects related to bicarbonate therapy, a relevant aspect given its role in correcting acidosis. A more thorough consideration of these aspects would have strengthened the overall depth and applicability of the review, providing a more nuanced understanding of the intricate relationship between acid stress, insulin action, and therapeutic interventions such as bicarbonate therapy.

We have added a section on side effects of bicarbonate therapy.  In addition, this paper is part of a special edition, and we reference another author in this edition who has addressed this topic in greater detail

Round 2

Reviewer 1 Report

Comments and Suggestions for Authors

The authors have taken the reviewers comments into consideration and have significantly improved the paper.

Reviewer 2 Report

Comments and Suggestions for Authors

Greetings

The authors have implemented substantial revisions in the manuscript, potentially positioning the paper for acceptance for publication.

Comments on the Quality of English Language

English language is OK